# Inorganic Polyphosphate Triggers NLRP3 Inflammasome and Promotes the Epithelial-to-Mesenchymal Transition and Migration of Colorectal Cancer Cells Through TRPM8 Receptor

**DOI:** 10.3390/ijms26167743

**Published:** 2025-08-11

**Authors:** Valentina Arrè, Maria Principia Scavo, Rossella Donghia, Francesco Dituri, Camilla Mandorino, Marco Cassotta, Anna Ancona, Francesco Balestra, Leonardo Vincenti, Fabrizio Aquilino, Giuseppe Pettinato, Gianluigi Giannelli, Roberto Negro

**Affiliations:** 1Personalized Medicine Laboratory, National Institute of Gastroenterology “S. de Bellis”, IRCCS Research Hospital, Via Turi 27, Castellana Grotte, 70013 Bari, Italy; francesco.dituri@irccsdebellis.it (F.D.); camilla.mandorino@irccsdebellis.it (C.M.); marco.cassotta@irccsdebellis.it (M.C.); 2Laboratory of Molecular Medicine, National Institute of Gastroenterology “S. de Bellis”, IRCCS Research Hospital, Via Turi 27, Castellana Grotte, 70013 Bari, Italy; maria.scavo@irccsdebellis.it (M.P.S.); francesco.balestra@irccsdebellis.it (F.B.); 3Data Science Unit, National Institute of Gastroenterology “S. de Bellis”, IRCCS Research Hospital, Via Turi 27, Castellana Grotte, 70013 Bari, Italy; rossella.donghia@irccsdebellis.it; 4Core Facility Biobank, National Institute of Gastroenterology “S. de Bellis”, IRCCS Research Hospital, Via Turi 27, Castellana Grotte, 70013 Bari, Italy; anna.ancona@irccsdebellis.it; 5Unit of Surgery, Department of Surgery Sciences, National Institute of Gastroenterology “S. de Bellis”, IRCCS Research Hospital, Via Turi 27, Castellana Grotte, 70013 Bari, Italy; leonardo.vincenti@irccsdebellis.it (L.V.); fabrizio.aquilino@irccsdebellis.it (F.A.); 6Division of Gastroenterology, Department of Medicine, Beth Israel Deaconess Medical Center, Harvard Medical School, 330 Brookline Avenue, Boston, MA 02215, USA; gpettina@bidmc.harvard.edu; 7Scientific Direction, National Institute of Gastroenterology “S. de Bellis”, IRCCS Research Hospital, Via Turi 27, Castellana Grotte, 70013 Bari, Italy; gianluigi.giannelli@irccsdebellis.it

**Keywords:** inorganic polyphosphate, NLRP3 inflammasome, TRPM8, colorectal cancer, epithelial–mesenchymal transition, migration

## Abstract

Inorganic polyphosphate (iPolyP) is a ubiquitous molecule composed of a variable number of orthophosphate units. Recent studies have highlighted its involvement in colorectal cancer (CRC) cell proliferation. However, further investigations are needed to elucidate its role in CRC cell progression and migration, as well as its influence on the tumor microenvironment. This study focuses on the inorganic polyphosphate (iPolyP)/transient receptor potential cation channel subfamily M member 8 (TRPM8) axis and its impact on CRC progression. To investigate these issues, western blotting, fixed and live cells immunofluorescence, 2D and 3D cell culture on CRC-patient derived tissues, ELISA, and wound healing assays were performed. Our results show that inorganic polyphosphate induces the expression of epithelial-to-mesenchymal transition (EMT) markers in CRC cells. Furthermore, the iPolyP/TRPM8 axis indirectly promotes tumor growth through activation of the Nucleotide-binding oligomerization domain, Leucine-rich Repeat and Pyrin domain-containing protein 3 (NLRP3) inflammasome in immune cells, leading to increased levels of the pro-inflammatory cytokine interleukin-1β (IL-1β) in the tumor microenvironment (TME), thereby advancing CRC. These findings suggest that targeting the iPolyP/TRPM8 pathway may be a promising strategy to inhibit CRC progression and metastasis.

## 1. Introduction

Colorectal cancer (CRC) is among the major public health concerns worldwide, with an incidence of approximately 10% of all cancer cases [1]. Gradually accumulating genetic or epigenetic alterations in healthy colonic epithelial cells underpin the transition from adenomas to invasive adenocarcinomas [2]. However, while the onset and development of CRC requires genetic modifications that mostly fall into three distinct classes of aberrations, namely chromosomal instability, microsatellite instability, and the cytosine-phosphate-guanine (CpG) island methylation phenotype [3,4,5,6,7,8,9], invasive, or metastatic, these properties follow the acquisition of mesenchymal traits from epithelial cells, a process called the epithelial–mesenchymal transition (EMT) [10]. By conferring aggressiveness, the EMT triggers apical–basal polarity disruption, tight junction dissolution, as well as cytoskeletal rearrangements, thus providing an infiltration and migratory ability to CRC cells [10,11,12,13,14,15,16,17,18,19,20,21,22]. Extracellular stimuli from the tumor microenvironment (TME) affect the complex molecular network that initiate and support the EMT. Particularly relevant in this context are the Transforming Growth Factor beta 1/Suppressor of Mothers against Decapentaplegic (TGF-β1/SMAD), Wingless/Integrated (Wnt)/β-catenin, and Neurogenic locus notch homolog protein 1 (Notch 1) pathways [23]. Not least, independent groups have shown that a key role in EMT progression and CRC invasion is mediated by the pro-inflammatory cytokines, such as interleukin-1β (IL-1β), released within the TME [24,25,26]. Notoriously, the CRC microenvironment is characterized by elevated levels of several pro-inflammatory cytokines, among which IL-1β, considered the major mediator of inflammation, stands out and fuels tumor invasion through immunosuppressive activities [27,28]. Bioactive IL-1β originates as a direct consequence of inflammasome activation in the innate immune cells [29]. Inflammasomes are supramolecular cytosolic structures found in all innate immune cells, which auto-assemble into micrometer-size complexes upon endogenous or exogenous insults and trigger the inflammatory response. Among the several known mammalian inflammasomes, the NLRP3 inflammasome is the best studied. After sensing and binding the threat, monomers of Nucleotide-binding oligomerization domain, Leucine-rich Repeat and Pyrin domain-containing protein 3 (NLRP3) organize into a wheel-shaped multimeric scaffold that signals to an adaptor protein, named apoptosis-associated speck-like protein containing a Caspase activation and recruitment domain (ASC), which, in turn, activates pro-Caspase-1, forming the core of the inflammasome. Active Caspase-1 cleaves pro-IL-1β, pro-IL-18, and gasdermin D (GSDMD) into their mature forms (IL-1β, IL-18, and N-terminus GSDMD, respectively). While the N-terminus GSDMD generates plasma membrane pores, allowing the release of IL-1β and IL-18 into the bloodstream, concomitantly an inflammatory form of cell death, called pyroptosis, occurs. Morphologically denominated as a “speck” or punctum, the active NLRP3 inflammasome colocalizes with the centrosome, also known as the microtubule-organizing center (MTOC), located in the perinuclear area [30]. Aberrant NLRP3 inflammasome activation has been reported to contribute, alongside other multiple risk factors, to the onset of several inflammation-driven diseases, including cancer [31,32,33]. Given the influence on CRC pathophysiology of the inflammasome-dependent and/or -independent EMT, intensive research is now underway. It has been recently reported that the pro-inflammatory molecule, named inorganic polyphosphate (iPolyP), is upregulated in neoplastic CRC tissues compared to the corresponding normal counterpart and contributes to the development and progression of CRC via its binding receptor called transient receptor potential cation channel subfamily M (melastatin) member 8 (TRPM8) [34]. Being composed by hundreds of orthophosphates linked together by adenosine triphosphate (ATP)-like bonds, iPolyP is considered an optimal source of energy with roles in disparate pathophysiological processes, including inflammation-driven diseases [35,36,37,38,39], tumorigenesis [40], tumor metastasis [41,42], and cellular proliferation [43,44]. Both iPolyP and the receptor TRPM8 are gathering attention in CRC pathophysiology nowadays, because while the former might be described as food additive, deriving from the intestinal microbiota or released by cellular organelles within the TME and promote a pro-inflammatory niche favorable for tumor growth [45], the latter has been associated with poor prognosis in CRC subjects, where it has been found to be overexpressed [46]. Through in vitro and ex vivo experimental approaches, the aim of our study was firstly to investigate whether the iPolyP/TRPM8 axis is involved in the EMT program. Additionally, considering its inflammatory properties, as well the abundance of high IL-1β in the serum of CRC subjects, we wished to rule out the potential role of iPolyP in the activation of the NLRP3 inflammasome in the CRC context. Thus, we tested the expression of several well-characterized EMT markers, such as Epithelial cadherin, Neural cadherin [47],Vimentin [48], phospho-Cofilin [49], Matrix Metalloprotease-2 [50], Alpha Smooth Muscle Actin [51], Type 1 Collagen [52], Connective Tissue Growth Factor [53], Fibronectin [54], and Filamentous Actin [55] in the presence or absence of iPolyP and TRPM8 inhibitor in a CRC cell line and CRC cell line-derived spheroids, as well as in CRC-patient derived organoids. Moreover, using the THP-1 human monocyte cell line, we investigated the possible influence of the iPolyP/TRPM8 axis on NLRP3 inflammasome activation. Together, our findings provide important insights into the involvement of the iPolyP/TRPM8 axis in CRC progression and could potentially pave the way for the development of novel anticancer agents as supplements to conventional chemotherapy.

## 2. Results

### 2.1. The iPolyP/TRPM8 Signaling Axis Promotes the Expression of the Epithelial to Mesenchymal Transition Markers in Colorectal Cancer Cells

It is believed that metastatic spread relies on genetic defects and the EMT program underpinning cellular plasticity. To determine the role of iPolyP on EMT onset, we treated human colorectal adenocarcinoma cell line Caco-2 and SW620 with iPolyP for 48 h and tested the expression level of EMT protein markers such as Alpha Smooth Muscle Actin (α-SMA) and Neural cadherin (N-cad) (Figure 1A and Appendix A), as well as Vimentin (Vim), phospho-Cofilin (phCFL), Metallopeptidase-2 (MMP2), Type 1 Collagen (COL1A1), and Connective Tissue Growth Factor (CTGF) (Appendix A), which resulted in enhanced compared to untreated (UT) cells, while no such difference was observed in the control cells HCEC-1CT (Figure 1A, Appendix A). Concomitantly, E-Cadherin expression was dampened in Caco-2 and SW620 upon treatment with iPolyP (Figure 1A, Appendix A). In addition, immunofluorescence analysis showed an increment of Fibronectin and Filamentous Actin (F-actin) in iPolyP-treated cancer cells compared to HCEC-1CT (Appendix A). Pharmacological inhibition of the TRPM8 channel re-established the resting condition (Figure 1A, Appendix A). Lastly, upon iPolyP administration, HCEC-1CT-, Caco-2-, and SW620-derived spheroids showed an α-SMA phenotype comparable to that detected in 2D culture, as mentioned above, with a complete recovery of the baseline level following treatment with the TRMP8 inhibitor (Appendix A). Similarly, patient-derived CRC organoids challenged with iPolyP underwent a significant increase in Vimentin (Vim), MMP2, α-SMA, COL1A1, and CTGF, which was dampened by blocking the TRPM8 receptor (Figure 1B, and Appendix A). Overall, these results indicate the involvement of the iPolyP/TRPM8 signaling axis in the development of the EMT program in the CRC context.

### 2.2. iPolyP Stimulates Colorectal Cell Migration via TRPM8 Receptor

Cancer progression is characterized by an increased cell motility and invasiveness. To assess the migratory activity of the iPolyP/TRPM8 axis on CRC cells, we performed wound healing assays on Caco-2 cells, compared with HCEC-1CT. The presence of iPolyP encouraged the complete sealing of both sides by Caco-2 cells, whereas no effect was observed in HCEC-1CT cells. Moreover, TRPM8 blockage prevented the migratory ability of iPolyP (Figure 2A,B). Thus, all together these results demonstrate a role of the iPolyP/TRPM8 pathway in supporting CRC cell migration.

### 2.3. iPolyP Triggers the NLRP3 Inflammasome Activation via TRPM8 Receptor

It is well-established nowadays that patients with colorectal cancer have high levels of circulating cytokines, such as IL-1β, responsible, in turn, for fostering the invasiveness of CRC. In line with the literature [26], we detected high levels of IL-1β in the serum of 10 subjects with CRC as compared to healthy individuals (Appendix A), suggesting, therefore, an underlying upregulation of the inflammasome complex. iPolyP is considered a pro-inflammatory molecule which contributes to shape the tumor immune microenvironment in several different types of cancers, including CRC, although the molecular mechanism has not yet been defined. Hence, we firstly explored whether iPolyP could be, at least partially, responsible for the production, maturation, and/or secretion of IL-1β through the inflammasome NLRP3, often misregulated in the CRC context [56]. Similarly to Caco-2 and SW620, human monocyte THP-1 cells express the TRPM8 receptor (Appendix A), which makes them suitable for the study of a possible role of the iPolyP/TRPM8 axis in NLRP3 inflammasome priming or activation. Strikingly, iPolyP-primed THP-1 cells showed enhanced cytosolic pro-IL-1β, seen as protein or mRNA, whose level was analogous to those observed in lipopolysaccharide (LPS) primed samples (Figure 3A,B), so far the best standardized first-step signal commonly used to trigger the TLR4-NLRP3 axis in monocytes and macrophages. Moreover, the presence of the TRPM8 inhibitor prevented the synthesis of pro-IL-1β (Figure 3A,B), thus assigning a role for iPolyP as a novel and unexpected player in the NLRP3 inflammasome priming pathway through the TRPM8 receptor. Consequently, iPolyP-primed THP-1 cells, followed by ATP stimulus, displayed a similar activation threshold to LPS-primed and Nigericin- or ATP-activated THP-1, used as positive controls, in terms of the canonical readouts normally explored in the inflammasome field, such as the release of mature IL-1β in the culture medium (Figure 3C), the cleavage of pro-Caspase-1 into its active form (p20 fragment) (Figure 3D), the formation of the so-called perinuclear “speck” or punctum (Figure 3E,F and Appendix A), as well as pyroptotic cell death (Figure 3G). A normal physiology of THP-1 cells, characterized by the presence of large electron-transparent autophagic vacuoles upon NLRP3 engagement, was observed under iPolyP/ATP stimulation. The presence of vacuoles was suppressed when the stimulation was combined with the TRPM8 inhibitor (Figure 3H). We next performed time-lapse video-microscopy experiments on stably expressing pro-IL-1β-mNeonGreen (mNG) (Excitation, 506 nm; Emission, 517 nm) and ASC-mScarlet-I (Excitation, 569 nm; Emission, 593 nm) THP-1, and stimulated them in the presence or absence of the TRPM8 inhibitor. Specks appearance was in line with the time-window previously reported in literature for canonical NLRP3 inflammasome activation (roughly 10–15 min) (Appendix A). However, no speck was detected in the presence of the TRPM8 inhibitor. All together, these results delineate, for the first time, a mechanistic interpretation underpinning the onset of the iPolyP/TRPM8-mediated pro-inflammatory niche within the CRC microenvironment.

### 2.4. IL-1β Derived from the iPolyP/NLRP3 Inflammasome Axis Promotes the Expression of Epithelial-to-Mesenchymal Transition Markers and Migration of Colorectal Cancer Cells

Following iPolyP/TRPM8-mediated NLRP3 inflammasome activation, we sought to outline the potential role of bioactive IL-1β, highly represented in the serum of CRC-affected subjects, in the development of the disease. Thus, conditioned media derived from iPolyP/ATP-induced NLRP3 inflammasome activation in THP-1 cells, were applied to recipient HCEC-1CT, Caco-2, and SW620 cells. The expression level of α-SMA and N-Cadherin in Caco-2 and SW620 cells was enhanced upon NLRP3 inflammasome activation compared to those observed in untreated cells, or in cells treated with IL-1β receptor antagonist. No detectable changes were observed in control HCEC-1CT cells. Conversely, E-Cadherin expression level was dampened compared to untreated control. To ascribe this phenotype specifically to IL-1β, the TRPM8 and NLRP3 inflammasome pathways were pharmacologically inhibited, thus recovering α-SMA, N-Cadherin, and E-Cadherin basal level (Figure 4A,B and Appendix A). Treatment of Caco-2 and SW620 cells with purified recombinant IL-1β recapitulated similar outcomes (Appendix A). As a direct consequence, Caco-2 cells displayed a significantly higher migratory rate when cultured with iPolyP + ATP compared to IL-1RA-enriched conditioned medium from THP-1 monocytes. The presence of TRPM8 and NLRP3 inhibitors restored the migratory rate observed in untreated samples (Figure 5A,B). Likewise, Caco-2 cells treated with IL-1β cytokine showed scratch closure within 24 h (Appendix A). These results proved an indirect role of the iPolyP/TRPM8 axis in EMT program, altering the expression of markers associated with mesenchymal traits, as well as CRC cell motility.

## 3. Discussion

Despite significant efforts to develop new therapeutic strategies, CRC remains one of the deadliest cancers in humans. This is largely due to the complex interplay of factors, such as age, environment, and genetic predisposition, which underpin each stage of tumorigenesis [57,58,59]. The EMT is a cellular process that plays a crucial role in the development and progression of CRC, thus now considered one of the major targets for preventing neoplastic cells from acquiring invasive traits. The EMT enables the transformation of polarized epithelial cells, characterized by tight junctions and apical–basal polarity, into motile mesenchymal cells with enhanced invasive properties, thus contributing to tumor heterogeneity as well as to the onset of niches in distant organs [60]. Consequently, cancer cells display stem-like characteristics that confer resistance to conventional therapeutic approaches. Multiple extracellular signaling pathways, synergistically or independently, converge toward the EMT. These include agents like: TGF-β1, which promotes the level of several transcription factors, like Snail family transcriptional repressor 1 (SNAIL), Twist Family BHLH Transcription Factor 1 (Twist1), and Zinc finger E-box-binding homeobox 1/2 (Zeb1/2); Epidermal Growth Factor (EGF); Hypoxia, which upregulates genes associated with the mesenchymal phenotype; the Wnt Signaling pathway [61]; and inflammatory cytokines, such as IL-1β [62].

### 3.1. iPolyP Induces the EMT Program to Enhance CRC Cell Migration

With its ATP-like bonds, accumulating evidence is contextualizing the inorganic polyphosphate in the cancer background as an energy supplier [40]. iPolyP has been enzymatically linked to two main sources, bacterial and human, that have different lengths. While bacterial iPolyP consists of 100 to 1000 units, and is synthesized by polyphosphate kinase 1 (ppk1), human-derived iPolyP averages around 60 to 100 phosphate residues of unknown enzymatic origin. It has been recently shown that inorganic polyphosphatase enzyme (ppx1), upregulated in CRC and known to hydrolyze long iPolyP chains into smaller chains, can activate the phosphatidylinositol 3-kinase/Protein kinase B (PI3K/AKT) pathway [63], which exhibited pivotal regulatory tasks in the EMT development [64], although the evidence remains indirect. In this study we demonstrate, through in vitro and ex vivo approaches, a direct involvement of the iPolyP/TRPM8 axis in the EMT process, which ultimately confers migratory properties to colorectal cancer cells, as well as a novel and unexpected involvement in the inflammatory pathway through the induction and secretion of the NLRP3 inflammasome-derived IL-1β, further enhancing CRC cell migration. These conclusions are based on the following evidences, summarized in Figure 4A: (i) iPolyP directly upregulates EMT markers, such as N-cadherin, Vimentin, phospho-CFL, MMP2, α-SMA, COL1A1, CTGF, Fibronectin, and F-actin, while downregulating E-cadherin, by interacting with the TRPM8 receptor; (ii) the iPolyP/TRPM8 axis governs CRC cell migration; (iii) iPolyP is able to shape the tumor microenvironment by priming the NLRP3 inflammasome, which is then fully activated by ATP, in macrophages; (iv) NLRP3-derived IL-1β induces the expression of EMT markers, which further sustain cell migration, conferring an additional, indirect, role of iPolyP in CRC spread (Figure 6).

### 3.2. iPolyP Indirectly Promotes CRC Progression Through Activation of the NLRP3 Inflammasome

Canonical NLRP3 inflammasome activation, a pathway specifically found in innate immune cells, requires two consecutive steps. The priming step (signal 1) is generally mediated by the engagement of the Toll-like receptor (TLR) on the macrophage surface, which recognizes structurally conserved microbial molecules (such as lipopolysaccharide, LPS, from Gram-negative bacteria) and leads to NF-kB-mediated upregulation of intracellular levels of NLRP3, pro-IL-1β, and pro-IL-18, whose initial concentrations are inadequate to initiate the assembly of NLRP3 in resting conditions [65]. Following signal 1, the activation step (signal 2), is provided by a broad range of pathogens-associated molecular patterns (PAMPs) or damage-associated molecular patterns (DAMPs), which include particulate matter, extracellular ATP, and pore-forming toxins. When paired, the two signals allow the oligomerization of NLRP3, Caspase-1 activation as well as the maturation of the executor proteins Gasdermin D, IL-1β, and IL-18 into their bioactive, mature forms [66]. Here, we identified a parallel mechanism, existing in innate immune cells, which involves the highly abundant molecule iPolyP and its matched receptor TRPM8, thus mimicking the well-known priming pathway. This novel signal 1, in combination with ATP, whose concentration is significantly enhanced within extracellular spaces and interstitial fractions in CRC [67], licenses the maturation and secretion of IL-1β at levels comparable to those seen upon LPS/ATP treatment. Hence, although our findings point out, for the first time, a direct and indirect involvement of the iPolyP/TRPM8 signaling axis in CRC growth and spread, more studies, including those involving in vivo approaches, are needed. Ongoing CRC animal models are, therefore, currently running in our laboratory, aiming to confirm our above-described findings. In addition, we will apply these findings to other cancer types which might help to classify iPolyP/TRPM8-sensitive/insensitive neoplasms. These results also open up new opportunities for therapeutic intervention. The TRPM8 inhibitor AMTB is currently under evaluation in early-phase clinical trials for other malignancies, such as pancreatic cancer. This suggests a potential for drug repurposing strategies targeting TRPM8 in CRC as well. Moreover, the enzymatic machinery responsible for iPolyP biosynthesis in eukaryotic cells remains unidentified, representing a completely novel target for drug development. Identifying the enzymes involved in iPolyP production and turnover could allow the design of novel inhibitors capable of modulating this pro-metastatic and pro-inflammatory pathway. In this context, a combinatorial approach targeting both iPolyP synthesis and TRPM8 receptor engagement may provide a synergistic strategy to limit CRC progression and metastasis. All together, these results uncover a novel, functional axis in the CRC context, shedding light on new directions for study and paving the way for the development of new therapeutic strategies for CRC patients. Further investigation using TRPM8-deficient Apc Min/+ mouse model of CRC is currently underway in our laboratory to corroborate these results. Additionally, we will perform experiments to test the presence of iPolyP in body fluids or stool samples. Early diagnosis of CRC typically relies on invasive methods with several systematic limitations and requires specialized skills. However, advancements have been made with liquid biopsy screening, which avoids invasive procedures and allows monitoring of the disease’s response to therapy. Therefore, by promoting proliferation, spread, and a pro-inflammatory environment in the CRC context, iPolyP could potentially serve as a novel non-invasive biomarker for early detection and disease monitoring through liquid biopsy. This could significantly reduce the reliance on invasive biopsy methods, offer long-term benefits, and improve healthcare outcomes, ultimately enhancing patients’ lives and having a significant social impact.

## 4. Materials and Methods

### 4.1. Patients Samples

Patients provided written informed consent to the collection of tissue specimens, as well as of 20 sera, including 10 derived from normal individuals and 10 from CRC affected patients, under Prot. No. 379/C.E. of 16 September 2020 of the Local Ethics Committee “Gabriella Serio” IRCCS Istituto Tumori “Giovanni Paolo II”, Bari, Italy. Tissue specimens and serum samples were provided by the Core Facility Biobanca of IRCCS “S. de Bellis, Castellana Grotte, Bari, Italy”. The diagnosis of colorectal cancer was confirmed histologically. Samples collected in the operating room were sectioned and freshly processed for the experiments.

### 4.2. Cell Culture and Reagents

Human colorectal adenocarcinoma Caco-2 and SW620 cells were purchased from the American Tissue Culture Collection (ATCC, Manassas, VA, USA; Cat. No.: HTB-37 and CCL-227, respectively). Human Colonic Epithelial Cells 1 transduced with Cyclin-Dependent Kinase 4 (CDK4) and Telomerase HCEC-1CT cells were purchased from Evercyte GmbH (Vienna, Austria; Cat. No.: CkHT-039-0229). Human embryonic kidney 293T (HEK293T) were purchased from the American Tissue Culture Collection (ATCC, Manassas, VA, USA; Cat. No.: CRL-3216). Human monocytic THP-1 cells, isolated from peripheral blood from an acute monocytic leukemia patient, were purchased from the American Tissue Culture Collection (ATCC, Manassas, VA, USA; Cat. No.: TIB-202). Caco-2, SW620 and HEK293T cells were grown in Dulbecco’s Modified Eagle’s medium (DMEM) (Thermo Fisher Scientific, Waltham, MA, USA; Cat. No.: 11965092), supplemented with 10% fetal bovine serum (FBS) (Thermo Fisher Scientific, Waltham, MA, USA; Cat. No.: A5256701), 1 mM Sodium Pyruvate (Thermo Fisher Scientific, Waltham, MA, USA; Cat. No: 11360039), 25 mM 4-(2-hydroxyethyl)-1-piperazineethanesulfonic acid (HEPES) (Thermo Fisher Scientific, Waltham, MA, USA; Cat. No.: 15630056), and 100 U/mL Antibiotic-Antimycotic (Thermo Fisher Scientific, Waltham, MA, USA; Cat. No.: 15240062). Human Colonic Epithelial Cells 1 transduced with CDK4 and Telomerase (HCEC-1CT) were grown in ColoUp medium ready to use (Evercyte GmbH, Vienna, Austria; Cat. No.: MHT-039), supplemented with 100 U/mL Antibiotic-Antimycotic (Thermo Fisher Scientific, Waltham, MA, USA; Cat. No.: 15240062). THP-1 cells were maintained in Roswell Park Memorial Institute medium (RPMI, Thermo Fisher Scientific, Waltham, MA, USA; Cat. No.: 11875093), supplemented with 10% FBS, 100 U/mL Antibiotic-Antimycotic (Thermo Fisher Scientific, Waltham, MA, USA; Cat. No.: 15240062) and 0.05 mM 2-mercaptoethanol (Sigma-Aldrich, St. Louis, MO, USA; Cat. No.: M6250-100 mL). All cell lines were maintained in a humidified atmosphere at 37 °C with 5% CO_2_. Cells were passaged and medium was changed every other day. Experiments were performed with cells at passage number 2–7.

### 4.3. Immunoblotting Assay

HCEC-1CT, Caco-2, and SW620 cell lines were seeded into six-well plates (Corning, New York, NY, USA; Cat. No.: 3516) at a density of 0.5 × 10^6^ cells/well in 2 mL of complete cell culture medium. Serum-deprived seeded cells were treated with 0.5 µM of sodium phosphate glass (iPolyP) (Sigma-Aldrich, St. Louis, MO, USA; Cat. No.: S4379-500mg), or with 10 µM of TRPM8 receptor inhibitor N-(3-Aminopropyl)-2-[(3-methylphenyl)methoxy]-N-(2-thienylmethyl)benzamide (AMTB) hydrochloride (Santa Cruz Biotechnology, Dallas, TX, USA; Cat. No.: sc-361103), or in combination for 72 h. Additionally, HCEC-1CT, Caco-2, and SW620 cells were treated with 10 ng/mL of purified human recombinant IL-1β (Thermo Fisher Scientific, Waltham, MA, USA; Cat. No.: 200-01B-10UG), with 100 ng/mL of IL-1β receptor antagonist (IL-1RA) (Thermo Fisher Scientific, Waltham, MA, USA; Cat. No.: 200-01RA), or in combination for 72 h. Dimethyl sulfoxide (DMSO) (Sigma-Aldrich, St. Louis, MO, USA; Cat. No.: D8418-100mL) was added to the control cells. Pharmacological inhibition of the iPolyP/TRPM8 axis was performed by adding TRPM8 inhibitor to the HCEC-1CT, Caco-2 and SW620 cell lines. THP-1 cells were employed for NLRP3 inflammasome studies. THP-1 cells were seeded into six-well plates at a density of 2 × 10^5^ cells/mL in 2 mL of complete cell culture medium and treated overnight with 300 ng/mL phorbol myristate acetate (PMA) (Sigma-Aldrich, St. Louis, MO, USA; Cat. No.: P8139-5MG). The following day, the priming step was performed by adding 1 µg/mL of lipopolysaccharides (LPS) (Sigma-Aldrich, St. Louis, MO, USA; Cat. No.: L4524-5MG), considered as positive control, or 0.5 µM iPolyP, to the cells for 4 h. Pharmacological inhibition of the priming step of NLRP3 inflammasome was performed by adding 10 µM of TRPM8 inhibitor for 4 h. NLRP3 inflammasome activation was triggered by adding 5 mM of adenosine 5′-triphosphate (ATP) disodium salt hydrate (Sigma-Aldrich, St. Louis, MO, USA; Cat. No.: FLAAS-1VL), or 20 µM of Nigericin sodium salt (Sigma-Aldrich, St. Louis, MO, USA; Cat. No.: N7143). Pharmacological inhibition of NLRP3 inflammasome activation step was performed by adding 0.1 µM of the NLRP3 direct inhibitor MCC950 (Sigma-Aldrich, St. Louis, MO, USA; Cat. No.: 5.38120) 1 h before the activation step. Besides, THP-1 were used to assess the role of inflammasome-derived IL-1β on the expression of EMT markers in HCEC-1CT, Caco-2, and SW620 cell. Specifically, THP-1 cells were treated with iPolyP + ATP to induce NLRP3 inflammasome activation, or left untreated (UT); inflammasome inhibition was performed with the addition of TRPM8 or NLRP3 specific inhibitor; IL-1β blockage was carried with the addition of IL-1RA (Thermo Fisher Scientific, Waltham, MA, USA; Cat. No.: 200-01RA), 1 h before the activation step. Conditioned media were then applied to serum-deprived cultured HCEC-1CT, Caco-2, and SW620 cell for 48 h. Following incubation/activation, adherent cells were detached using 0.05% Trypsin-EDTA (Thermo Fisher Scientific, Waltham, MA, USA; Cat. No.: 25300-054), centrifuged at 1100 rpm for 5 min at 4 °C and washed with 1x sterile Dulbecco’s phosphate-buffered saline 1 (1x DPBS) (Thermo Fisher Scientific, Waltham, MA, USA; Cat. No.: 14190-094), twice. Dried pellets were frozen at −80 °C or resuspended and lysed in 200 µL of T-PERTM Tissue Protein Extraction Reagent (Thermo Fisher Scientific, Waltham, MA, USA; Cat. No.: 78510) supplemented with Halt™ Protease and Phosphatase Inhibitor Single-Use Cocktail, Ethylenediaminetetraacetic Acid (EDTA)-Free (100x) (Thermo Fisher Scientific, Waltham, MA, USA; Cat. No.: 78443), for immunoblotting analysis. Cellular lysates were incubated on ice for 30 min and vortexed every 10 min. Samples were then centrifuged at 16,000 rpm at 4 °C for 20 min to clarify and precipitate insoluble debris. Total extracted proteins were assayed to measure concentrations using the Bio-Rad protein assay dye reagent concentrate (Bio-Rad Laboratories, Hercules, CA, USA; Cat. No.: 5000006EDU). Then, proteins were mixed with 4 x Laemmli Sample Buffer (Bio-Rad Laboratories, Hercules, CA, USA; Cat. No.: 1610747) and 10% of β-mercaptoethanol (Sigma-Aldrich, St. Louis, MO, USA; Cat. No.: M6250-100mL) and denatured at 95 °C for 5 min. A total of 25 µg of proteins were loaded onto precast polyacrylamide 4–20% gels (Bio-Rad Laboratories, Hercules, CA, USA; Cat. No.: 4568094), subsequently blotted on a polyvinylidene fluoride (PVDF) membrane (Bio-Rad Laboratories, Hercules, CA, USA; Cat. No.: 1704156) using the trans-blot turbo transfer system (Bio-Rad Laboratories, Hercules, CA, USA; Cat. No.: 1704150). Membranes were blocked using Pierce™ Protein-Free Blocking Buffer (Thermo Fisher Scientific, Waltham, MA, USA; Cat. No.: 37571) for 1 h and stained overnight with primary antibodies. The next day membranes were washed three times with 1x Tris Buffered saline (1x TBS) (Bio-Rad Laboratories, Hercules, CA, USA; Cat. No.: 1706435 diluted in ddH_2_O to reach 1x)/Polyoxyethylene (20) sorbitan monolaurate (TWEEN20) (Sigma-Aldrich, St. Louis, MO, USA; Cat. No.: P9416-100mL) incubated for 1 h with the respective horseradish peroxidase-conjugated secondary antibodies. Proteins were detected using the Clarity Max Western Enhanced Chemiluminescence (ECL) Substrate (Bio-Rad Laboratories, Hercules, CA, USA; Cat. No.: 1705062) and the signals were obtained using the Chemidoc MP Imaging System (Bio-Rad Laboratories, Hercules, CA, USA; Cat. No.: 1708280). The following primary antibodies were used: anti-TRPM8, 1:1000 (Thermo Fisher Scientific, Waltham, MA, USA; Cat. No.: MA5-35474); anti-E-cadherin, 1:1000 (Antibodies, Stockholm, Sweden; Cat. No.: A250830); anti-N-cadherin, 1:1000 (Abcam, Cambridge Biomedical Campus, Cambridge, UK; Cat. No.: ab18203); anti-Vimentin, 1:1000 (Cell Signaling Technology, Danvers, MA, USA; Cat. No.: 57415); anti-phospho-Cofilin (Ser3), 1:500 (Cell Signaling Technology, Danvers, MA, USA; Cat. No.: 33115); anti-Cofilin, 1:500 (Cell Signaling Technology, Danvers, MA, USA; Cat. No.: 33185); anti-MMP2, 1:500 (Abcam, Cambridge Biomedical Campus, Cambridge, UK; Cat. No.: ab86607); anti-α-SMA, 1:500 (Merck, Rahway, NJ, USA; Cat. No.: A5228); anti-COL1A1 1:1000 (Cell Signaling Technology, Danvers, MA, USA; Cat. No.: 911445); anti-CTGF, 1:500 (Abcam, Cambridge Biomedical Campus, Cambridge, UK; Cat. No.: ab6992); anti-TRPM8, 1:1000 (Thermo Fisher Scientific, Waltham, MA, USA; Cat. No.: MA5-35474); anti-pro-IL-1β, 1:1000 (Abcam, Cambridge Biomedical Campus, Cambridge, UK; Cat. No.: ab226918); anti-pro-Caspase-1, 1:1000 (Cell Signaling Technology, Danvers, MA, USA; Cat. No.: 3866S); anti-GAPDH, 1:1000 (Santa Cruz Biotechnology, Dallas, TX, USA; Cat. No.: sc-47724). GAPDH was used as loading controls. The following secondary antibodies were employed: Anti-rabbit IgG, HRP-linked Antibody, 1:2000 (Cell Signaling Technology, Danvers, MA, USA; Cat. No.: 7074S); Goat anti-Mouse IgG (H+L)-HRP Conjugate (Bio-Rad Laboratories, Hercules, CA, USA; Cat. No.: 1706516).

### 4.4. CRC Tumor Organoids

CRC specimens from surgically resected tumor tissue were washed three times with 1x sterile DPBS (Thermo Fisher Scientific, Waltham, MA, USA; Cat. No.: 14190-094) and digested with collagenase/hyaluronidase mixture (Stemcell Technologies, Vancouver, BC, Canada; Cat. No.: 07912), diluted in Hanks’ Balanced Salt Solution (HBSS) solution (with CaCl_2_ and MgCl_2_, Thermo Fisher Scientific, Waltham, MA, USA; Cat. No.: 14025-050) for 5 h under gentle rocking at 37 °C. A single cell suspension was obtained and cells were embedded in matrigel matrix basement membrane (Corning, New York, NY, USA; Cat. No.: 356231), and cultured in IntestiCult-SF (ICT-SF) medium (Stemcell Technologies, Vancouver, BC, Canada; Cat. No.: 100-0340) medium diluted 1:2 in Advanced DMEM/F-12 (Thermo Fisher Scientific, Waltham, MA, USA; Cat. No.: 12634-010), supplemented with N-2 (Thermo Fisher Scientific, Waltham, MA, USA; Cat. No.: 17502-048), B27 supplement (Thermo Fisher Scientific, Waltham, MA, USA; Cat. No.: 12587-010), 0.01% bovine serum albumin (BSA), 100 U/mL Antibiotic-Antimycotic (Thermo Fisher Scientific, Waltham, MA, USA; Cat. No.: 15240062), and HEPES (Thermo Fisher Scientific, Waltham, MA, USA; Cat. No.: 15630056). Medium was renewed every two days until the organoids were fully developed (7–10 days). Mature organoids were split using TrypLE Select dissociation agent (Thermo Fisher Scientific, Waltham, MA, USA; Cat. No.: A12177-01) according to [54] and the suspended cells obtained were re-cultured at a lower density in matrigel.

### 4.5. Immunofluorescence of Tumor Cells Derived from CRC Tumor Organoids

Cells obtained from organoids splitting were seeded at low density on microscopy slides (Nunc Lab-Tek II, Merck Life Sciences, Milan, Italy; Cat. No.: 154534) previously coated with matrigel diluted 1:10 in 1x sterile DPBS (Thermo Fisher Scientific, Waltham, MA, USA; Cat. No.: 14190-094), and left to attach and grow for at least 7 days in ICT-SF medium until the formation of large islets in a humidified incubator set to 37 °C and 5% CO_2_. Cells were treated with Dimethyl sulfoxide (DMSO, indicated as untreated, UT) (Sigma-Aldrich, St. Louis, MO, USA; Cat. No.: D8418-100 mL), 0.5 µM iPolyP, TRPM8 inhibitor 10 µM, or iPolyP + TRPM8 inhibitor for 48 h, and fixed with 4% paraformaldehyde solution (PFA) (Sigma-Aldrich, St. Louis, MO, USA; Cat. No.: P6148-500G) in 1x sterile DPBS (Thermo Fisher Scientific, Waltham, MA, USA; Cat. No.: 14190-094) for 15 min at 4 °C, permeabilized with 0.15% Triton X-100 (Sigma-Aldrich, St. Louis, MO, USA; Cat. No.: 101895731) for 7 min at room temperature, blocked with 2% BSA (Sigma-Aldrich, St. Louis, MO, USA; Cat. No.: A7030-100G) in 1x sterile DPBS (Thermo Fisher Scientific, Waltham, MA, USA; Cat. No.: 14190-094) for 1 h at room temperature, and incubated overnight with primary antibodies (anti-Vimentin, 1:1000; anti-MMP2, 1:1000; anti-α-SMA, 1:500; anti-COL1A1, 1:1000; anti-CTGF; antibodies details are reported in the Immunoblotting section). The following day, cells were washed three times with 1x sterile DPBS (Thermo Fisher Scientific, Waltham, MA, USA; Cat. No.: 14190-094), incubated for 1 h at room temperature with fluorophore-conjugated secondary antibodies for the detection of the antigens of interest and subjected to microscopy analysis. The following secondary antibodies were used: Goat anti-Rabbit IgG (H+L) Highly Cross-Adsorbed Secondary Antibody, Alexa Fluor™ Plus 488, 1:500 (Thermo Fisher Scientific, Waltham, MA, USA; Cat. No.: A-32731TR); Goat anti-Mouse IgG (H+L) Highly Cross-Adsorbed Secondary Antibody, Alexa Fluor™ 488, 1:500 (Thermo Fisher Scientific, Waltham, MA, USA; Cat. No.: A-11029). Images were collected and analyzed as described in the Cell lines immunofluorescence and confocal microscopy section.

### 4.6. Immunofluorescence of Cell-Line-Derived Spheroids

1 × 10^3^ Caco-2, SW620, and HCEC-1CT cells were seeded into 3D low attachment 96-well cell culture plates (Corning, New York, NY, USA; Cat. No.: 4520) to obtain 3D cell-line spheroids with 100 µL of growth medium in each well. Cells were kept in culture in the above-described conditions. After seeding, all cell lines were treated with 0.5 µM iPolyP, with 10 µM TRPM8 inhibitor, or both, while untreated cells were used as controls (untreated, UT). Spheroid cultures were observed at 24 and 96 h and maintained in a humidified incubator set to 37 °C and 5% CO_2_. After 96 h, cells were treated with Dimethyl sulfoxide (DMSO, indicated as untreated, UT) (Sigma-Aldrich, St. Louis, MO, USA; Cat. No.: D8418-100 mL), 0.5 µM iPolyP, TRPM8 inhibitor 10 µM, or iPolyP + TRPM8 inhibitor for 48 h, and fixed with 4% PFA (Sigma-Aldrich, St. Louis, MO, USA; Cat. No.: P6148-500G) in 1x sterile DPBS (Thermo Fisher Scientific, Waltham, MA, USA; Cat. No.: 14190-094) for 15 min at 4 °C, permeabilized with 0.15% Triton X-100 (Sigma-Aldrich, St. Louis, MO, USA; Cat. No.: 101895731) for 7 min at room temperature, blocked with 2% BSA (Sigma-Aldrich, St. Louis, MO, USA; Cat. No.: A7030-100G) in 1x sterile DPBS (Thermo Fisher Scientific, Waltham, MA, USA; Cat. No.: 14190-094) for 1 h at room temperature, and incubated overnight with α-SMA antibody. The following day cells were washed three times with 1x sterile DPBS (Thermo Fisher Scientific, Waltham, MA, USA; Cat. No.: 14190-094), incubated for 1 h at room temperature with fluorophore-conjugated secondary antibody: Goat anti-Rabbit IgG (H+L) Highly Cross-Adsorbed Secondary Antibody, Alexa Fluor™ Plus 488, 1:500 (Thermo Fisher Scientific, Waltham, MA, USA; Cat. No.: A-32731TR). Nuclei were stained using PureBlu 4′,6-diamidino-2-phenylindole (DAPI) (Bio-Rad Laboratories, Hercules, CA, USA; Cat. No.: 1351303). Spheroids were then subjected to microscopy analysis. Images were collected and analyzed as described in the Cell lines immunofluorescence and confocal microscopy section.

### 4.7. Cell Line Immunofluorescence and Confocal Microscopy Assay

2 × 10^5^/mL HCEC-1CT, Caco-2, SW620, and THP-1 cells were grown in 35 mm petri dishes, no. 1.5 coverglass (MatTek, Ashland, MA, USA; Cat. No: P35G-1.5-14-C). Serum-deprived HCEC-1CT, Caco-2, and SW620 cells were treated with 0.5 µM iPolyP, or with 10 µM TRPM8 inhibitor or both for 48 h. THP-1 cells were treated overnight with 300 ng/mL phorbol myristate acetate (PMA) (Sigma-Aldrich, St. Louis, MO, USA; Cat. No.: P8139-5MG). The following day, the priming step was performed by adding 1 µg/mL of lipopolysaccharides (LPS) (Sigma-Aldrich, St. Louis, MO, USA; Cat. No.: L4524-5MG), considered as positive control, or 0.5 µM iPolyP, to the cells for 4 h. Pharmacological inhibition of the priming step of the NLRP3 inflammasome was performed by adding 10 µM of TRPM8 inhibitor for 4 h. NLRP3 inflammasome activation was triggered by adding 5 mM of adenosine 5′-triphosphate (ATP) disodium salt hydrate (Sigma-Aldrich, St. Louis, MO, USA; Cat. No.: FLAAS-1VL) or 20 µM of Nigericin sodium salt (Sigma-Aldrich, St. Louis, MO, USA; Cat. No.: N7143). Pharmacological inhibition of the NLRP3 inflammasome was performed by adding 0.1 µM of the NLRP3 direct inhibitor MCC950 (Sigma-Aldrich, St. Louis, MO, USA; Cat. No.: 5.38120) 1 h before the activation step. Pharmacological blockage of IL-1β was assessed by adding 100 ng/mL of IL-1RA (Thermo Fisher Scientific, Waltham, MA, USA; Cat. No.: 200-01RA) to the media, 1 h before the activation step. Similarly to immunoblotting experiments, UT or stimulated THP-1-derived conditioned media were applied to cultured HCEC-1CT, Caco-2, and SW620. Fixative, permeabilization, and blocking buffers were prepared in 1x sterile DPBS (Thermo Fisher Scientific, Waltham, MA, USA; Cat. No.: 14190-094). Cells were fixed with 4% PFA (Sigma-Aldrich, St. Louis, MO, USA; Cat. No.: P6148-500G) for 30 min at 4 °C and then washed twice using 1x sterile DPBS. Permeabilization was performed for 7 min at room temperature using 0.15% Triton X-100 (Sigma-Aldrich, St. Louis, MO, USA; Cat. No.: 101895731) diluted in 1x DPBS. Washing was performed to remove permeabilization buffer. Cells were then blocked for 1 h at room temperature using blocking buffer (3% BSA) (Sigma-Aldrich, St. Louis, MO, USA; Cat. No.: A7030-100G)/1x sterile DPBS). Cells were incubated with primary (overnight) and secondary (1 h) antibodies. Nuclei were stained using PureBlu DAPI (Bio-Rad Laboratories, Hercules, CA, USA; Cat. No.: 1351303). In between, extensive washing steps were performed to remove unbound antibodies and stains. The following primary antibodies were employed: anti-α-SMA 1:200 (Merck, Rahway, NJ, USA; Cat. No.: A5228); anti-E-cadherin, 1:200 (Antibodies, Stockholm, Sweden; Cat. No.: A250830); anti-N-cadherin 1:200 (Abcam, Cambridge Biomedical Campus, Cambridge, UK; Cat. No.: ab18203); anti-Fibronectin, 1:1000 (Thermo Fisher Scientific, Waltham, MA, USA; Cat. No.: MA5-11981); rhodamine phalloidin, for the F-actin detection, 1:250 (Thermo Fisher Scientific, Waltham, MA, USA; Cat. No.: R415); anti-ASC (anti-ASC/TMS1 (E1E3I)), 1:1000 (Cell Signaling Technology, Danvers, MA, USA; Cat. No.: 13833S). The following secondary antibodies were used: Goat anti-Rabbit IgG (H+L) Highly Cross-Adsorbed Secondary Antibody, Alexa Fluor™ Plus 488, 1:500 (Thermo Fisher Scientific, Waltham, MA, USA; Cat. No.: A-32731TR); Goat anti-Mouse IgG (H+L) Highly Cross-Adsorbed Secondary Antibody, Alexa Fluor™ 488, 1:500 (Thermo Fisher Scientific, Waltham, MA, USA; Cat. No.: A-11029). Images were collected with a Nikon Ti2 inverted laser-scanning confocal microscope equipped with Plan Apo 20x (0.75 numerical aperture) in bright-field and analyzed with Nikon Imaging Software (NIS)-Elements software (version 5.11.01) and Fiji software (version 2.16.0). Prior to acquisition, cells were placed in the microscope CO_2_ chamber. All images were collected with a confocal Yokogawa spinning-disk (Musashino, Tokyo, Japan) on a Nikon Eclipse Ti2 inverted microscope (Nikon Instruments Inc., Melville, New York, NY, USA), equipped with Plan Fluor 40x (0.60 numerical aperture) lens. Images were acquired with a Hamamatsu ORCA ER cooled CCD camera (Hamamatsu Photonics, Chūō-ku, Hamamatsu City, Shizuoka, Japan) controlled with NIS-Elements software (version 5.11.01). For time-lapse experiments, images were collected using an exposure time of 700 milliseconds. At each time point, z-series optical sections were collected with a step size of 1 µM at the indicated time intervals. Gamma, brightness, and contrast were adjusted on displayed images (identically for comparative image sets) using NIS-Elements software (version 5.11.01). The Perfect Focus System was kept running for continuous maintenance of focus. DMEM without phenol red (Thermo Fisher Scientific, Waltham, MA, USA; Cat. No.: 21063-045) or RPMI without phenol red (Thermo Fisher Scientific, Waltham, MA, USA; Cat. No.: 11835063) were used during image acquisition.

### 4.8. Wound Healing Assay

HCEC-1CC, Caco-2, and SW620 cells were seeded in a 12-well plate (Corning, New York, NY, USA; Cat. No.: 3513) in 10% FBS medium and grown to reach an 80–90% confluence. Then, a scratch was made on the cell monolayer using a p200 pipette tip. The plate was washed with 1x sterile DPBS to remove the debris and markings were created on the outer bottom of the plate with a tip marker, to be used as reference points. Each cell line was incubated with medium without FBS, containing either 0.5 µM iPolyP, 10 µM TRPM8 inhibitor, or in combination. Additionally, HCEC-1CT, Caco-2, and SW620 cells were treated with 10 ng/mL of recombinant human IL-1β (Thermo Fisher Scientific, Waltham, MA, USA; Cat. No.: 200-01B-10UG), with 100 ng/mL of IL-1RA (Thermo Fisher Scientific, Waltham, MA, USA; Cat. No.: 200-01RA), or in combination. THP-1 cells were treated with iPolyP + ATP, with TRPM8, or in combination. Conditioned media were then applied to the cultured HCEC-1CT, Caco-2, and SW620 cell. Following the scratch, images were acquired at 0 and 24 h using a confocal microscope Nikon Eclipse Ti2 (Nikon Instruments Inc., Melville, New York, NY, USA) in bright field microscopy equipped with a Plan Fluor 4x (0.10 numerical aperture) lens and Fiji software (version 2.16.0) was used to measure the scratched area. Cell migratory ability for wound-healing was assessed by applying the following formula: [(wound area at 0 h) − (wound area at 24 h)]/(wound area at 0 h).

### 4.9. Detection of Cytokines by ELISA

The detection kit for human IL-1β (Abcam, Cambridge Biomedical Campus, Cambridge, UK; Cat. No.: ab214025) was used at the specified temperature and conditions according to the manufacturer’s instructions. Briefly, 50 μL serum derived from Normal and Pathological individuals were placed into 96-well plates, added with an additional 50 μL of antibody cocktail. The plates were then incubated for 2 h at room temperature on a plate shaker set to 400 rpm. After extensive washing, 100 μL of TMB development solution were added to each well for 10 min in the dark on a plate shaker set to 400 rpm. Finally, 100 μL of stop solution were added to each well before recording the OD at 450 nm.

### 4.10. Propidium Iodide Fluorescence Cell Death Assay

After inflammasome priming by LPS or iPolyP for 4 h and activation by LPS + nigericin, LPS + ATP, LPS + ATP + AMTB, iPolyP + ATP, iPolyP + ATP + AMTB, THP-1 cells were collected, alongside the untreated condition, and washed twice in 1x sterile DPBS. Pellets were then re-suspended in 2 µg/mL propidium iodide (PI) (ImmunoChemistry Technologies LLC, Davis, CA, USA; Cat. No.: 638). The percentage of cells which took up PI was measured by flow cytometry (Navios) (Beckman Coulter, Brea, CA, USA; Cat. No.: B83535) and the results were analyzed with FlowJo version 10 Software (FlowJo LLC, Ashland, OR, USA).

### 4.11. Cloning

Human ASC (a gift from Eicke Latz (Addgene plasmid #41840; http://n2t.net/addgene:41840; RRID:Addgene_41840, accessed on 1 July 2025) was subcloned into the third-generation lentiviral expression vector, plasmid lentivirus (pLV)-mScarlet (a gift from Pantelis Tsoulfas (Addgene plasmid #159172; http://n2t.net/addgene:159172; RRID:Addgene_159172, accessed on 1 July 2025), plasmid between SalI-HF and EcoRV-HF (New England Biolabs, Ipswich, MA, USA; Cat. No.: R0138M and R3195M, respectively) restriction sites using the following primers (Thermo Fisher Scientific, Waltham, MA, USA; Cat. No.: 10629186): forward primer of 5′-ATACGTCGACGGATCTATGGGGCGC-3′ and reverse primer of 5′-TTCAGATATCTAACTCGATGGTAGC-3′.

### 4.12. Generation Stable Cell Lines

To generate stable cell lines, on day 0, lentivirus was produced using HEK293T cells by cotransfecting 1 mg of plasmid Lentiviral Vector (pLV) containing the gene: 1. IL-1β: pLV-mTurquoise2-IL-1β-mNeonGreen was a gift from Hao Wu (Addgene plasmid #166783; http://n2t.net/addgene:166783; RRID:Addgene_166783, accessed on 1 July 2025) or 2. pLV mScarlet-I-ASC with 750 ng of psPAX2 packaging plasmid (a gift from Didier Trono (Addgene plasmid #12260; http://n2t.net/addgene:12260; RRID:Addgene_12260, accessed on 1 July 2025), and 250 ng of pMD2.G envelope plasmid (a gift from Didier Trono (Addgene plasmid #12259; http://n2t.net/addgene:12259; RRID:Addgene_12259, accessed on 1 July 2025). The transfected cells were incubated overnight. The following day (day 1), the medium was removed and the cells were replenished with 1 mL of fresh medium and incubated for another day. On day 2, the supernatant containing the virus was filtered using a 0.45-mm filter (Biosigma, Cona, Italy; Cat. No.: 051230) and directly used to infect THP-1 with a spinfection protocol to increase the efficacy. Spinfection was performed at 2500× *g* for 90 min at room temperature using 8 mg/mL polybrene (EMD Millipore Corp., Burlington, MA, USA; Cat. No.: TR-1003-G). Positive clones were selected by cell sorting and colonies were expanded from single clones. Positive clones were extensively validated by immunofluorescence microscopy.

### 4.13. Transmission Electron Microscopy

For transmission electron microscopy (TEM), THP-1 cells, following activation with ATP or nigericin activation, were processed for plastic embedding. Cells were first incubated in fixative for 1 h at room temperature. To prevent cellular shock and facilitate gentle fixation, a 2X fixative mixture was added to the culture medium at a 1:1 ratio in the dish containing the cells. This approach ensured optimal preservation of cellular structures for subsequent TEM analysis. Fresh fixative was prepared using 1.25% PFA (Sigma-Aldrich, St. Louis, MO, USA; Cat. No.: P6148-500G), 2.5% glutaraldehyde (Sigma-Aldrich, St. Louis, MO, USA; Cat. No.: 354400), and 0.03% picric acid (Sigma-Aldrich, St. Louis, MO, USA; Cat. No: P6744-1GA) in a 0.1 M sodium cacodylate buffer (Sigma-Aldrich, St. Louis, MO, USA; Cat. No.: 97068), pH 7.4. After fixation, the cells were washed three times in 0.1 M sodium cacodylate buffer. Subsequently, they were incubated with 1% osmium tetroxide/1.5% potassium ferrocyanide for 1 h at room temperature. Following this step, the cells were washed three times with water and then incubated in aqueous solution with 1% uranyl acetate (Electron Microscopy Science, Hatfield, PA, USA; Cat. No.: 541-09-3) for 30 min. This was followed by another three rounds of washing with water to ensure thorough removal of excess staining agents. Dehydration steps were performed twice in different grades of alcohol (70% ethanol for 15 min, 90% ethanol for 15 min, and 100% ethanol for 15 min). Samples were then placed in propyleneoxide (Sigma-Aldrich, St. Louis, MO, USA; Cat. No.: 110205) for 1 h and infiltration was performed with Epon mixed 1 + 1 with propyleneoxide for 3 h at room temperature. Samples were moved to the embedding mold filled with freshly mixed Epon and allowed to polymerize for 24 to 48 h at 60 °C. Ultrathin sections (~60 nm) were cut on a Reichert Ultracut-S microtome, placed on copper grids, and stained with lead citrate. The grids from the above-described electron microscopy procedures were examined using a JEOL Jem-1011 transmission electron microscope (JEOL U.S.A., Inc., Peabody, MO, USA) operating at an accelerating voltage of 100 kV. Images were acquired using an Olympus Quemesa Camera (11 Mpx) (Olympus, Shinjuku-ku, Tokyo, Japan) to ensure high-quality image capture.

### 4.14. Statistical Analysis

Parameters recorded are reported as mean and standard deviation (M ± SD). In total, 20 serums were collected, including 10 derived from normal individuals and 10 from CRC affected patients. Student’s *t*-test was used to compare two independent groups. Analysis of variance (ANOVA) was performed to assess differences among groups, and post-hoc analysis multiple pairwise comparisons, with Bonferroni adjustment to reduce the risk of Type I error. To test the null hypothesis of no association, the two-tailed probability level was set at < 0.05. The analyses were conducted with GraphPad Prism 10 (GraphPad Software, Boston, MA, USA).

## 5. Conclusions

The finding regarding the involvement of the iPolyP/TRPM8 axis in assisting the EMT program, along with the onset of a pro-inflammatory niche, could pave the way for developing novel therapeutic strategies against CRC that target TRPM8 activity. This could complement the currently approved conventional chemotherapy protocols and potentially improve their efficacy.

## Figures and Tables

**Figure 1 ijms-26-07743-f001:**
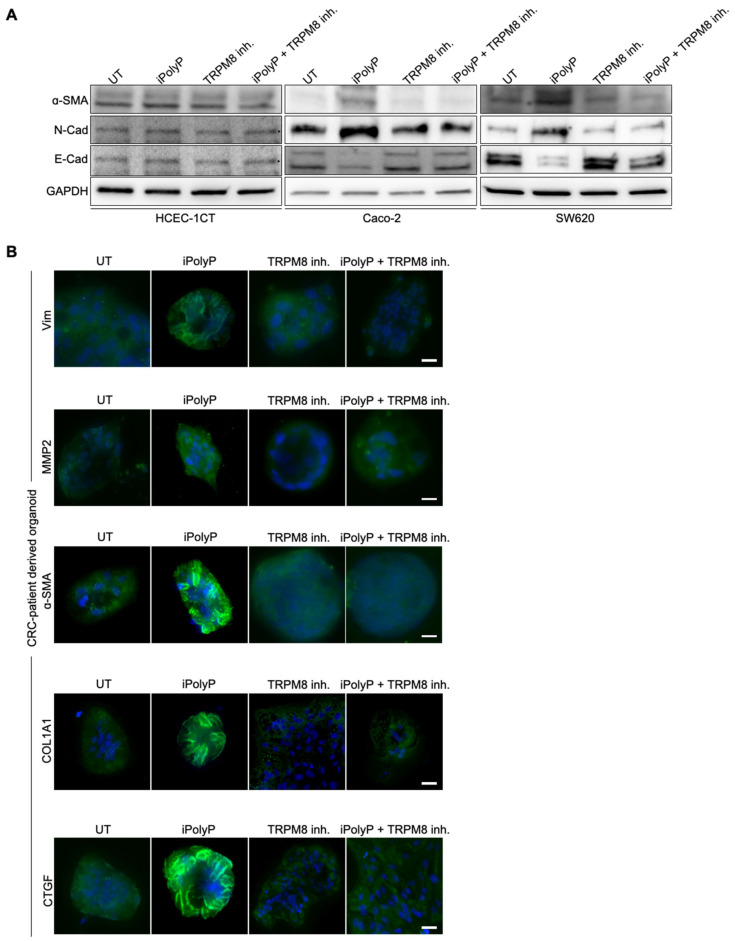
iPolyP induces expression of markers involved in the epithelial-to-mesenchymal transition. (**A**) Cellular extracts from HCEC-1CT, Caco-2, and SW620, treated with 0.5 µM iPolyP, 10 µM TRPM8 inhibitor, or both in combination for 48 h, were analyzed by immunoblotting for α-SMA, N-Cadherin, and E-Cadherin expression level. GAPDH was used as loading control. (**B**) Immunofluorescence on CRC patient’s-derived organoids, treated with 0.5 µM iPolyP, 10 µM TRPM8 inhibitor, or 0.5 µM iPolyP + 10 µM TRPM8 inhibitor 48 h to detect the expression of the above-mentioned proteins (green); blue represents nuclear staining by DAPI. Scale bar 100 µM. Images are representative of three independent experiments.

**Figure 2 ijms-26-07743-f002:**
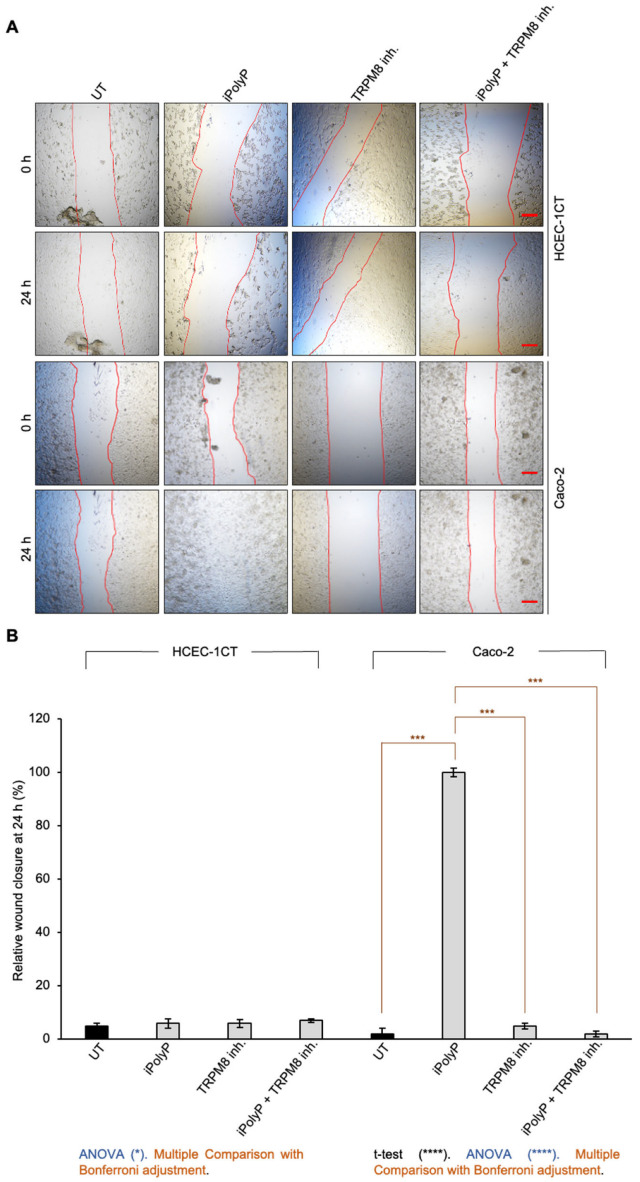
Effect of iPolyP-TRPM8 axis on cell migration. (**A**) Representative phase-contrast microscope images taken at 0 h and 24 h of a wound healing assay in HCEC-1CT and Caco-2 treated with 0.5 µM iPolyP, 10 µM TRPM8 inhibitor, as well as the combination of both. Red line delineates the cell-free region. Scale bar 500 µM. Images are representative of three independent experiments. (**B**) Quantification relative to panel A. Fold changes versus control, untreated (UT). Statistical analysis performed by Student’s *t*-test (iPolyP versus UT, **** *p* < 0.0001), ANOVA (in blue, * *p* < 0.05, **** *p* < 0.0001) and Post-hoc Multiple Comparisons Test with Bonferroni adjustment (in brown, *** *p* < 0.001). Parameters recorded are reported as mean and standard deviation (M ± SD).

**Figure 3 ijms-26-07743-f003:**
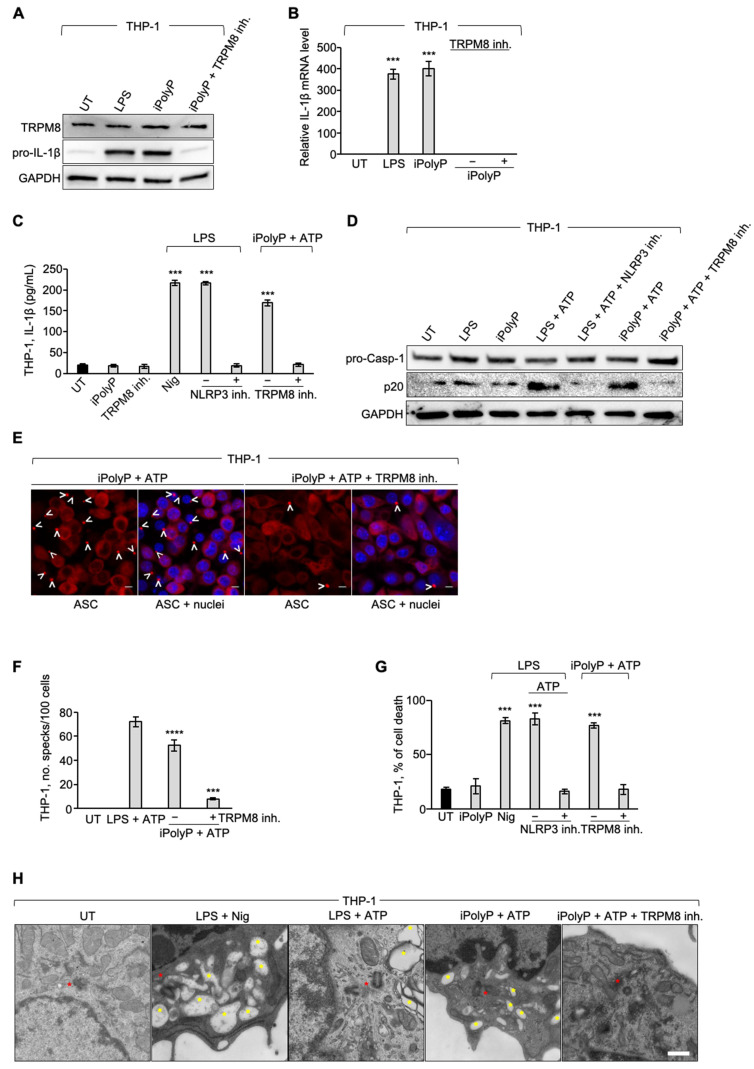
iPolyP triggers NLRP3 inflammasome priming in human monocyte. (**A**) Cellular extracts from THP-1 monocyte, treated with 1 µg/mL LPS, 0.5 µM iPolyP, or 0.5 µM iPolyP + 10 µM TRPM8 inhibitor for 4 h were analyzed by immunoblotting against pro-IL-1β and TRPM8, normalized on GAPDH. (**B**) Real-Time PCR on 4 h LPS-, iPolyP-, or iPolyP + TRPM8 inhibitor-treated THP-1 cells. Statistical analysis performed by Student’s *t*-test (experimental condition versus UT, *** *p* < 0.001). (**C**) Evaluation of IL-1β performed by ELISA on the supernatant of THP-1 cells treated with 0.5 µM iPolyP (4 h), 10 µM TRPM8 inhibitor (4 h), 1 µg/mL LPS (4 h) + 20 µM Nigericin (30 min), 1 µg/mL LPS (4 h) + 5 mM ATP (30 min), 1 µg/mL LPS (4 h) + 5 mM ATP (30 min) + 0.1 µM NLRP3 inhibitor (4 h), 0.5 µM iPolyP (4 h) + 5 mM ATP (30 min), or 0.5 µM iPolyP (4 h) + 5 mM ATP (30 min) + 10 µM TRPM8 inhibitor (4 h). Statistical analysis performed by Student’s *t*-test (experimental condition versus UT, *** *p* < 0.001). (**D**) Immunoblotting on THP-1 cells treated with 1 µg/mL LPS (4 h), 0.5 µM iPolyP (4 h), 1 µg/mL LPS (4 h) + 5 mM ATP (30 min), 1 µg/mL LPS (4 h) + 5 mM ATP (30 min) + 0.1 µM NLRP3 inhibitor (4 h), 0.5 µM iPolyP (4 h) + 5 mM ATP (30 min), or 0.5 µM iPolyP (4 h) + 5 mM ATP (30 min) + 10 µM TRPM8 inhibitor (4 h), and probed against pro-Caspase-1. Normalization was performed on GAPDH. (**E**) Representative confocal micrographs showing the inflammasome speck on fixed THP-1 cells treated with 1 µg/mL LPS (4 h) + 5 mM ATP (30 min), 0.5 µM iPolyP (4 h) + 5 mM ATP (30 min), or 0.5 µM iPolyP (4 h) + 5 mM ATP (30 min) + 10 µM TRPM8 inhibitor (4 h). White angle brackets indicate the presence of ASC specks (red). Blue represents nuclear staining by DAPI. Scale bar 10 µM. Images are representative of three independent experiments. (**F**) Speck quantification relative to panel (**E**). Statistical analysis performed by Student’s *t*-test (experimental condition versus UT, **** *p* < 0.0001; *** *p* < 0.001). (**G**) Cell death assay performed by Fluorescence-Activated Cell Sorting (FACS)-assisted PI staining on THP-1 cells treated with 0.5 µM iPolyP (4 h), 1 µg/mL LPS (4 h) + 20 mM Nigericin (30 min), 1 µg/mL LPS (4 h) + 5 mM ATP (30 min), 1 µg/mL LPS (4 h) + 5 mM ATP (30 min) + 0.1 µM NLRP3 inhibitor (4 h), 0.5 µM iPolyP (4 h) + 5 mM ATP (30 min), or 0.5 µM iPolyP (4 h) + 5 mM ATP (30 min) + 10 µM TRPM8 inhibitor (4 h). Statistical analysis performed by Student’s *t*-test (experimental condition versus UT, *** *p* < 0.001). (**H**) Representative electron microscopy micrographs of THP-1 cells treated with 1 µg/mL LPS (4 h) + 20 mM Nigericin (30 min), 1 µg/mL LPS (4 h) + 5 mM ATP (30 min), 0.5 µM iPolyP (4 h) + 5 mM ATP (30 min), or 0.5 µM iPolyP (4 h) + 5 mM ATP (30 min) + 10 µM TRPM8 inhibitor (4 h). Yellow asterisks denote the formation of vacuoles typical of NLRP3 inflammasome activation. Red asterisks denote the presence of the speck between centrioles, located near the nucleus. Scale bar 1 µM. Images are representative of three independent experiments. Fold changes versus control, untreated (UT). Parameters recorded are reported as mean and standard deviation (M ± SD).

**Figure 4 ijms-26-07743-f004:**
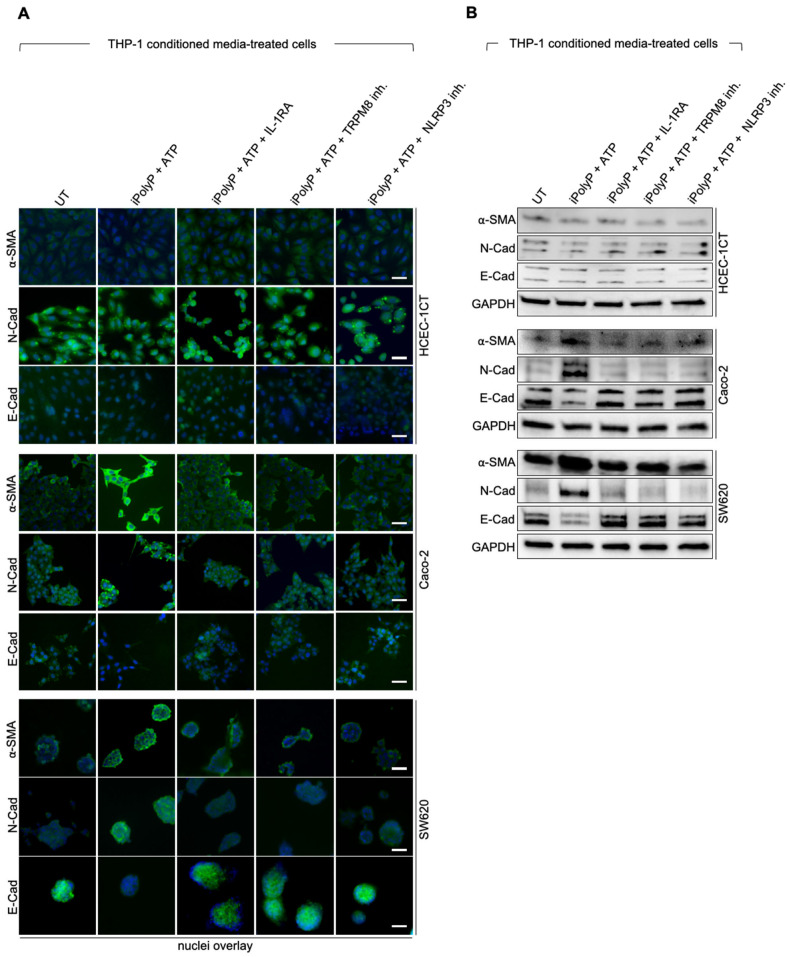
NLRP3 inflammasome-derived IL-1β modulates the expression of EMT markers in CRC cells. (**A**) Immunofluorescence images showing α-SMA, N-Cadherin, and E-Cadherin (green) on HCEC-1CT, Caco-2, and SW620 cultured with THP-1-derived conditioned media treated as follow: 0.5 µM iPolyP + 5 mM ATP, 0.5 µM iPolyP + 5 mM ATP + 100 ng/mL IL-1RA, 0.5 µM iPolyP + 5 mM ATP + 10 µM TRPM8 inhibitor, 0.5 µM iPolyP + 5 mM ATP + 0.1 µM NLRP3 inhibitor, or left untreated (UT) for 48 h. Blue represents nuclear staining by DAPI. Scale bar 10 µM. Images are representative of three independent experiments. (**B**) Cellular extracts from HCEC-1CT, Caco-2, and SW620, cultured with THP-1-derived conditioned media treated as follow: 0.5 µM iPolyP + 5 mM ATP, 0.5 µM iPolyP + 5 mM ATP + 100 ng/mL IL-1RA, 0.5 µM iPolyP + 5 mM ATP + 10 µM TRPM8 inhibitor, 0.5 µM iPolyP + 5 mM ATP + 0.1 µM NLRP3 inhibitor, or left untreated (UT) for 48 h were analyzed by immunoblotting for α-SMA, N-Cadherin, and E-Cadherin expression level (from top to bottom). GAPDH was used as loading control. Images are representative of three independent experiments.

**Figure 5 ijms-26-07743-f005:**
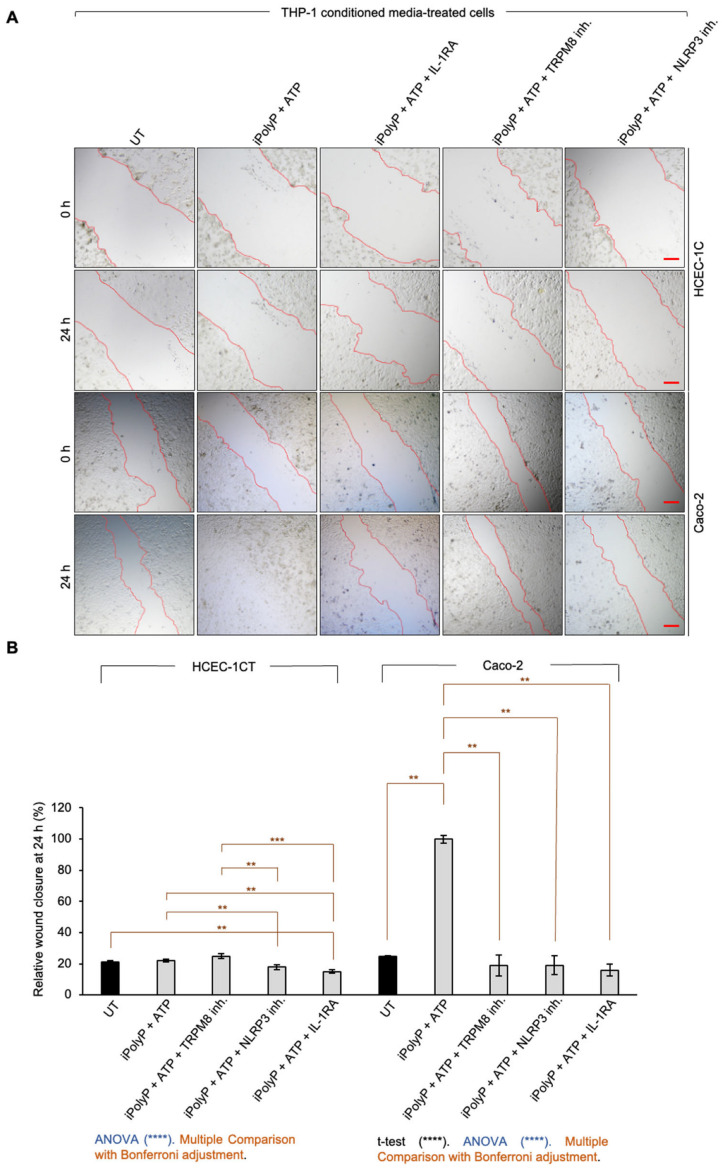
NLRP3 inflammasome-derived IL-1β triggers CRC cell migration. (**A**) Representative phase-contrast microscope images taken at 0 h and 24 h of a wound healing assay in HCEC-1CT and Caco-2 cultured with THP-1-derived conditioned media treated as follow: 0.5 µM iPolyP + 5 mM ATP, 0.5 µM iPolyP + 5 mM ATP + 100 ng/mL IL-1RA, 0.5 µM iPolyP + 5 mM ATP + 10 µM TRPM8 inhibitor, 0.5 µM iPolyP + 5 mM ATP + 0.1 µM NLRP3 inhibitor, or left untreated (UT). Red line delineates the cell-free region. Scale bar 500 µM. (**B**) Quantification relative to panel A. Fold changes versus control, untreated (UT). Statistical analysis performed by Student’s *t*-test (iPolyP + ATP versus UT, **** *p* < 0.0001), ANOVA (in blue, **** *p* < 0.0001), and Post-hoc Multiple Comparisons Test with Bonferroni adjustment (in brown, ** *p* < 0.01, *** *p* < 0.001). Parameters recorded are reported as mean and standard deviation (M ± SD).

**Figure 6 ijms-26-07743-f006:**
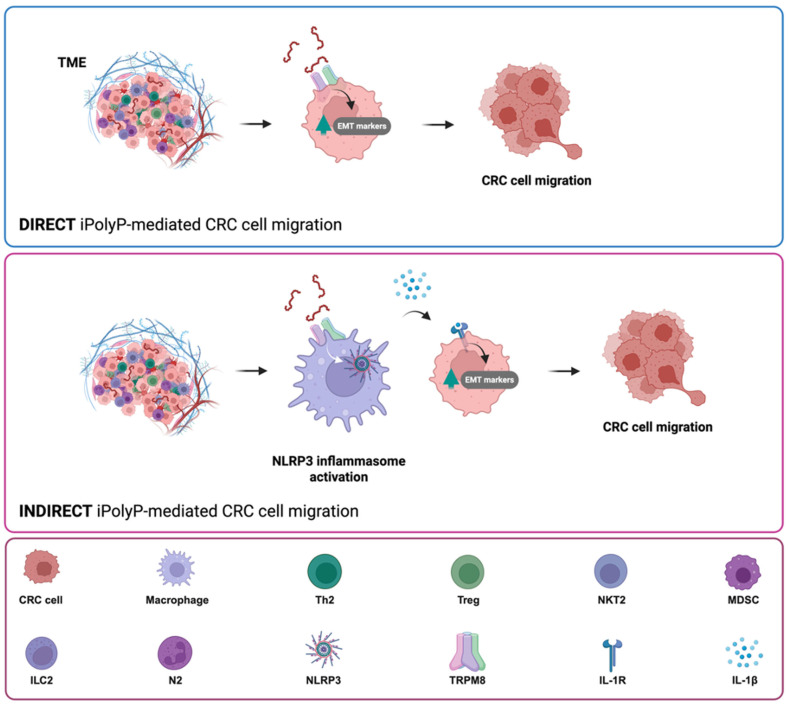
iPolyP triggers the expression of EMT markers and induces CRC cell migration through direct and indirect pathways. iPolyP, abundant within the tumor niche, enhances the expression of several EMT markers by signaling through TRPM8 receptor, which ultimately promotes CRC cell migration and invasiveness (direct iPolyP-mediated CRC cell migration) (**upper panel**). In addition, iPolyP/TRPM8 axis triggers the NLRP3 inflammasome activation in tumor-resident macrophages, leading to IL-1β secretion, which, in turn, fosters CRC cells migration (indirect iPolyP-mediated CRC cell migration) (**middle panel**). Legend is shown in the **lower panel**.

## Data Availability

The original raw data presented in the study are openly available at Appendix A.

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
