# Peer review of "Inorganic Polyphosphate Triggers NLRP3 Inflammasome and Promotes the Epithelial-to-Mesenchymal Transition and Migration of Colorectal Cancer Cells Through TRPM8 Receptor"

_ijms, 2025, doi:10.3390/ijms26167743_

Round 1
Reviewer 1 Report
Comments and Suggestions for Authors
The authors of the article “Inorganic polyphosphate triggers NLRP3 inflammasome and promotes the epithelial-to-mesenchymal transition and migration of colorectal cancer cells through TRPM8 receptor” present a well thought and concise study looking at the role of inorganic polyphosphates (iPolyP) in CRC progression and migration. The manuscript explores CRC EMT in a step-by-step manner that is easy to follow manner that validates its findings using inhibitors of key players to support their suggested role. All in all, I thoroughly enjoyed this manuscript.
Some minor points that need to be addressed are as follows:
- Figure 1: Please add what the green and blue colours in the row of images represent. This can be done with a small, coloured word in the top corner of one of the boxes or in the figure legend.
- Figure 3: Please add significance to graph F for -/+ iPolyP + ATP.
- Supplementary S6: Add LPS + ATP to quantification graph.
Author Response
The authors of the article “Inorganic polyphosphate triggers NLRP3 inflammasome and promotes the epithelial-to-mesenchymal transition and migration of colorectal cancer cells through TRPM8 receptor” present a well thought and concise study looking at the role of inorganic polyphosphates (iPolyP) in CRC progression and migration. The manuscript explores CRC EMT in a step-by-step manner that is easy to follow manner that validates its findings using inhibitors of key players to support their suggested role. All in all, I thoroughly enjoyed this manuscript.
We thank the Reviewer for his/her kinds words.
Some minor points that need to be addressed are as follows:
- Figure 1: Please add what the green and blue colours in the row of images represent. This can be done with a small, coloured word in the top corner of one of the boxes or in the figure legend.
- We thank the Reviewer for pointing this out. We have indicated the proper color information to all figure
legends, highlighted in yellow in the revised version of the manuscript.
- Figure 3: Please add significance to graph F for -/+ iPolyP + ATP.
- We agree with the Reviewer and have added the significance to the graph.
- Supplementary S6: Add LPS + ATP to quantification graph.
- We thank the Reviewer for the question. However, we would like to highlight that the quantification of
speck number upon LPS + ATP stimulation has already been shown in Figure 3F. In Supplementary
Figure S6, we use LPS + ATP as a positive control for another canonical inflammasome activator (LPS
+ Nigericin), allowing for a comparison between the two.

Reviewer 2 Report
Comments and Suggestions for Authors
The study presents a novel insight into the role of inorganic polyphosphate (iPolyP) in colorectal cancer (CRC) progression, particularly its impact on epithelial-to-mesenchymal transition (EMT) and migration. The exploration of the iPolyP/TRPM8 axis in CRC progression and its contribution to the inflammatory microenvironment via NLRP3 inflammasome activation is a significant and innovative aspect. The findings open the door for further therapeutic targeting in CRC.The experimental approach is thorough, utilizing various in vitro and ex vivo models, including CRC patient-derived organoids and spheroids, to study the iPolyP/TRPM8 signaling axis. However, the inclusion of additional in vivo validation could strengthen the claims and demonstrate the physiological relevance of the findings. The use of multiple CRC cell lines (Caco-2, SW620) and immune cell models (THP-1) is a solid approach, though more details on the specific conditions under which the assays were performed (e.g., passage number, medium composition) would improve reproducibility.
1.how does the iPolyP axis compare to other known pro-inflammatory factors in CRC?
2.The figures are generally well-organized, and the data is clearly presented. However, the legends for figures such as Figure 3 and Figure 4 could benefit from more detailed descriptions of the experimental conditions, such as specific drug concentrations and incubation times. For example, in Figure 3, specifying the time point of inflammasome activation could provide additional clarity.
3.The discussion section addresses the relevance of the findings well, particularly in the context of CRC metastasis. However, the manuscript could benefit from a more in-depth discussion on the potential clinical implications of targeting the iPolyP/TRPM8 axis in CRC. Are there any existing compounds that can inhibit this axis, or is it a novel target for drug development? Including a section on possible therapeutic approaches based on the findings would add value.
Author Response
The study presents a novel insight into the role of inorganic polyphosphate (iPolyP) in colorectal cancer (CRC) progression, particularly its impact on epithelial-to-mesenchymal transition (EMT) and migration. The exploration of the iPolyP/TRPM8 axis in CRC progression and its contribution to the inflammatory microenvironment via NLRP3 inflammasome activation is a significant and innovative aspect. The findings open the door for further therapeutic targeting in CRC. The experimental approach is thorough, utilizing various in vitro and ex vivo models, including CRC patient-derived organoids and spheroids, to study the iPolyP/TRPM8 signaling axis. However, the inclusion of additional in vivo validation could strengthen the claims and demonstrate the physiological relevance of the findings. The use of multiple CRC cell lines (Caco-2, SW620) and immune cell models (THP-1) is a solid approach, though more details on the specific conditions under which the assays were performed (e.g., passage number, medium composition) would improve reproducibility.
We thank the Reviewer for his/her insightful comments. In the “4.2. Cell culture and reagents” sub-section of Material and Methods section, we have included the following sentence, highlighted in turquoise: “Experiments were performed with cells at passage number 2-7”, page 18, line 435-436.
- How does the iPolyP axis compare to other known pro-inflammatory factors in CRC?
- We thank the Reviewer for the question. Our lab is currently investigating the molecular mechanism(s) that trigger inflammasome activation through engagement of the iPolyP/TRPM8 axis. According to the literature, most inflammatory pathways within the tumor microenvironment (TME) converge on the NF-κB transcription factor, culminating in its nuclear translocation and enhanced expression of pro-inflammatory genes. The iPolyP/TRPM8 axis could play a synergistic role, acting in parallel with previously described pathways, thereby amplifying NF-κB activation and reinforcing the inflammatory signaling cascade within the TME.
- The figures are generally well-organized, and the data is clearly presented. However, the legends for figures such as Figure 3 and Figure 4 could benefit from more detailed descriptions of the experimental conditions, such as specific drug concentrations and incubation times. For example, in Figure 3, specifying the time point of inflammasome activation could provide additional clarity.
- We thank the Reviewer for the note. We have now added the specific drug concentration in all the Figure legends, highlighted in turquoise.
- The discussion section addresses the relevance of the findings well, particularly in the context of CRC metastasis. However, the manuscript could benefit from a more in-depth discussion on the potential clinical implications of targeting the iPolyP/TRPM8 axis in CRC. Are there any existing compounds that can inhibit this axis, or is it a novel target for drug development? Including a section on possible therapeutic approaches based on the findings would add value.
- We thank the Reviewer for pointing this out. The TRPM8 inhibitor (AMTB), which we have extensively used to address the manuscript’s underlying rationale, has recently been included in various clinical trial programs for other types of cancer (e.g., pancreatic cancer), suggesting that its application may extend to additional malignancies, including colorectal cancer (CRC). Moreover, the enzymatic pathway responsible for iPolyP formation in eukaryotic cells remains unidentified. Elucidating this pathway could potentially broaden the landscape of therapeutic interventions against CRC.
We have now included in the Discussion section a short paragraph on the possible therapeutic approaches based on our findings, highlighted in turquoise: “These results also open up new opportunities for therapeutic intervention. The TRPM8 inhibitor AMTB, is currently under evaluation in early-phase clinical trials for other malignancies, such as pancreatic cancer. This suggests a potential for drug repurposing strategies targeting TRPM8 in CRC as well. Moreover, the enzymatic machinery responsible for iPolyP biosynthesis in eukaryotic cells remains unidentified, representing a completely novel target for drug development. Identifying the enzymes involved in iPolyP production and turnover could allow the design of novel inhibitors capable of modulating this pro-metastatic and pro-inflammatory pathway. In this context, a combinatorial approach targeting both iPolyP synthesis and TRPM8 receptor engagement may provide a synergistic strategy to limit CRC progression and metastasis.”, on page 17, line 374-383.
